# EXPLANATION-AWARE SOFT ENSEMBLE EMPOWERS LARGE LANGUAGE MODEL IN-CONTEXT LEARNING

## ABSTRACT

Large language models (LLMs) have shown remarkable capabilities in various natural language understanding tasks. With only a few demonstration examples, these LLMs can quickly adapt to target tasks without expensive gradient updates. Common strategies to boost such "in-context" learning ability are to ensemble multiple model decoded results and require the model to generate an explanation along with the prediction. However, these models often treat different class predictions equally and neglect the potential discrepancy between the explanations and predictions. To fully unleash the power of explanations, we propose EASE, an *Explanation-Aware Soft Ensemble* framework to empower in-context learning with LLMs. We design two techniques, explanation-guided ensemble, and soft probability aggregation, to mitigate the effect of unreliable explanations and improve the consistency between explanations and final predictions. Experiments on seven natural language understanding tasks and four varying-size LLMs demonstrate the effectiveness of our proposed framework.

## 1 INTRODUCTION

Recent advancements in Natural Language Processing (NLP) have witnessed the remarkable capabilities of Large Language Models (LLMs) (Brown et al., 2020; Tay et al., 2023; Chowdhery et al., 2022; Anil et al., 2023; Touvron et al., 2023; OpenAI, 2023). These LLMs can rapidly adapt to new tasks by learning only on a few input-output pairs (*a.k.a.* demonstrations) in context, without any gradient update (Wei et al., 2022a; Xie et al., 2022). Yet, beyond those demonstrations, a significant facet of human learning revolves around explanations. These explanations[1], typically in the form of a few keywords or sentences, reveal the underlying principles connecting the input and output (Zaidan et al., 2007; Narang et al., 2020). Consequently, the integration of free-text explanations into LLM prompting holds great potentials to further enhance in-context learning performance.

Recent studies have examined how to incorporate free-text explanations into LLM in-context learning scheme. For instance, the *Predict-then-Explain* pipeline (Lampinen et al., 2022) proposes to generate the explanation *after* making the prediction. Consequently, the predictions from LLM won't directly benefit from their corresponding explanations. In contrast, the *Explain-then-Predict* pipeline (also called "Chain-of-Thought") (Nye et al., 2021; Wei et al., 2022b) generates explanations *before* making predictions via greedy sampling. When the LLM-generated explanations are unreliable, predictions from this approach will be largely distracted and defective (Ye & Durrett, 2022). To mitigate this issue, Wang et al. (2023c) improves the "Chain-of-Thought" pipeline by first generating multiple predictions with different explanations using temperature sampling and then aggregating them via majority voting. However, this approach can be sub-optimal as (1) temperature sampling increases the inconsistency between generated explanations and their associated class predictions, and (2) majority voting treats different predictions associated with explanations of varying qualities equally. As a result, how to robustly leverage natural language explanations for empowering LLM in-context learning remains an open research question.

In this work, we present a novel **E**xplanation-**a**ware **S**oft **E**nsemble framework, named EASE, to assist LLM in-context learning with explanations. Our technique integrates explanations into the ensemble procedure and employs soft probability to mitigate discrepancies between explanations

---

[1]In this paper, we use the term 'explanations' and 'rationales' interchangeably.

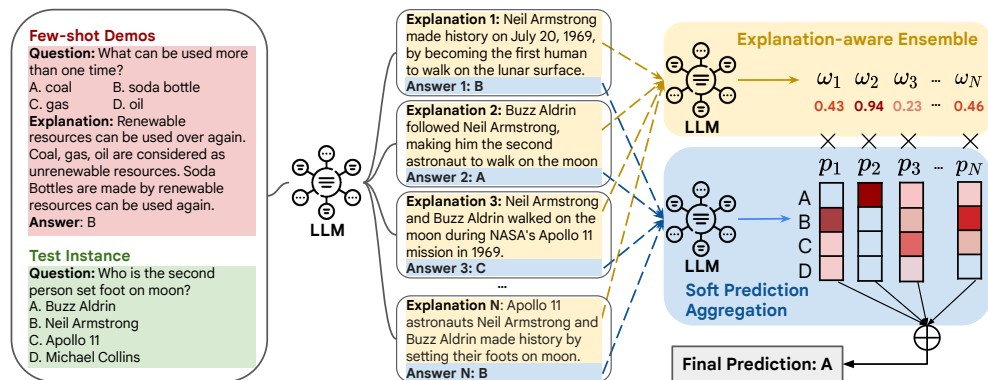

Figure 1: The overview of EASE framework.

and predictions. The key module of the EASE framework hinges upon the idea of weighted ensemble: As shown in Figure 1, instead of treating all predictions equally, we assign a score to each prediction based on the contextual relevance and inherent quality of its associated explanation, which will be used as a weight during the final ensemble stage. This explanation-aware ensemble stage is also realized with an LLM — after generating explanations and predictions using temperature sampling for each test instance, we prompt the LLM to weight all class predictions based on their associated explanations in an in-context manner. While the LLM offers great promise for the weighting purpose, it is crucial to provide sufficient *supervision signals* as demonstrations to guide the LLM scoring, yet the primary constraint for this step lies in the absence of *negative* explanations from few-shot demonstrations. To construct negative examples efficiently, we first use LLM to generate explanations for few-shot demonstrations, then select explanations associated with *incorrect predictions* as the negative samples. In this way, the LLM scorer can be readily applied to perform explanation-aware ensembling without any additional annotation.

Beyond explanation-aware ensembling, EASE incorporates an additional technique named *soft probability aggregation*, which helps to mitigate the *inconsistency* between explanations and predictions, given the sampling process may inevitably infuse noises into the final prediction. Specifically, it employs probabilities across various class-indicative verbalizers in place of the original one-hot predictions. This design, although conceptually simple, can effectively reduce the discrepancies between explanations and predictions and further improve the final predictions accuracy.

Our contributions can be summarized as follows:

- We propose the EASE framework to better facilitate in-context learning for large language models with natural language explanations.

- We design two techniques, namely explanation-aware ensemble and soft probability aggregation, to enable the model to focus on predictions associated with explanations of higher qualities while reducing the inconsistency between explanations and predictions.

- We conduct experiments on seven natural language understanding (NLU) datasets spanning between natural language inference (NLI) and question answering (QA), and our method outperforms previous state-of-the-art approaches using different LLMs as the backbone. Our analysis further justifies the advantages of using LLMs for explanation weighting to support correct answer candidates and leveraging soft probability aggregation to mitigate inconsistent predictions.

## 2 RELATED WORK

Two prevalent explanation types exist for interpreting NLP models: (1) *extraction-based explanations* that highlight important segments of the original input text (Zhang et al., 2016; DeYoung et al., 2020; Paranjape et al., 2020; Zhou et al., 2020; Yin & Neubig, 2022; Krishna et al., 2023) and (2) *free-form explanations* that craft prediction rationales directly using natural language text (Rajani et al., 2019; Sun et al., 2022; Wiegreffe et al., 2021; 2022; Wang et al., 2023a; Ludan et al., 2023). Beyond aiding in model interpretation, recent studies have demonstrated that these explanations can also enhance the few-shot learning capabilities of large language models. For example, (Bills et al., 2023) analyze the neuron activations to study factors in the text that trigger a neuron's activation.

(Krishna et al., 2023) leverage the extracted keywords tokens as additional input to assist LLM in-context learning. Wei et al. (2022b); Zelikman et al. (2022) propose to *prepend explanations* before the answers while Lampinen et al. (2022) suggest adding *post-answer explanations*. Given that these explanations are often derived during the LLM decoding stage and may contain noise, (Wang et al., 2023c; 2022) advocate for generating multiple candidate explanations with their respective predictions, followed by aggregating these predictions via majority voting. In our study, we focus on *free-form explanations* and explore how to better aggregating these predictions with explanations in a weighted ensemble. Using a bootstrapped LLM, we subsequently evaluate each explanation to enhance in-context learning outcomes.

Another line of research related to our study is automated explanation quality evaluation (Sun et al., 2022; Joshi et al., 2023; Wiegreffe et al., 2021; Chen et al., 2023a;b). Ye & Durrett (2022) utilize lexical features to measure the faithfulness of explanations without considering their semantics. Chen et al. (2021); Li et al. (2023b) leverage a NLI fine-tuned model to verify the explanations reliability. (Fu et al., 2023a; Liu et al., 2023; Qin et al., 2023; Jiang et al., 2023) also study how to use LLM to build a generic text quality scorers for generation and ranking tasks. These studies often rely on additional ground-truth labels and human annotations, making them less suitable when the labels for test instances are unknown. In contrast, our research focus more on effectively leveraging model-generated explanations to empower the LLM in-context learning performance.

## 3 METHOD

In this section, we first give a brief introduction to the problem definition. Then, we present our approach with two designs, namely explanation-aware ensemble and soft probability aggregation, with the goal of leveraging the generated explanations to improve the final prediction performance.

### 3.1 PROBLEM DEFINITION

In this task, we are given a LLM $\mathcal{M}$ parameterized by $\theta$, a set of few-shot demonstrations $\mathcal{D} = \{(x_i, e_i, y_i)\}_{i=1}^{K}$ on a target classification task[2], where $K$ is the number of demonstrations, $x_i, y_i$ are the input text and label for the $i$-th example, and $e_i$ is the corresponding ground-truth explanation. For each test example $x \in \mathcal{D}_{\text{test}}$, we aim to leverage $\mathcal{M}$ and $\mathcal{D}$ to predict its own label. Our primary goal is to improve the prediction accuracy for test examples.

### 3.2 RECAP OF SELF-CONSISTENCY PIPELINE FOR IN-CONTEXT LEARNING

Here we give a brief introduction to the self-consistency approach (Wang et al., 2023c). For each test example $x \in \mathcal{D}_{\text{test}}$, it first forms the prompt for few-shot demonstrations as $\mathcal{P} = \{\mathcal{T}, \text{shuffle}(\|_{i=1}^{K}(x_i, e_i, y_i))\}$, where $\mathcal{T}$ is the prompt template, and $\text{shuffle}(\|_{i=1}^{K}(x_i, e_i, y_i))$ is a permutation of $K$ demonstrations. Then, it generates $N$ candidate explanations together with predictions (denoted as $(e_j, p_j)$) via sampling from the LLM with non-zero temperature as

$$(e_j, p_j)_{j=1}^{N} \sim p_\theta(e, p \mid \mathcal{P}, x), \tag{1}$$

Finally, it aggregates these $N$ candidates into the final prediction via majority voting as

$$\widetilde{y} = \underset{y}{\text{argmax}} \sum_{j=1}^{N} \mathbb{I}(p_j = y). \tag{2}$$

Self-consistency enhances the standard explain-then-predict pipeline by utilizing multiple predictions derived from varied explanations. Despite its strong performance, through our examination, we've pinpointed two primary bottlenecks within the self-consistency pipeline, listed as follows:

- *Explanation-agnostic Ensembling*: Self-consistency uniformly weights all predictions and aggregates them via simple majority voting. This approach overlooks the variance in explanation quality, which can be problematic when certain predictions stem from flawed reasoning paths evident in poor-quality explanations.

- *Explanation-Prediction Inconsistency*: During its prediction phase, Self-consistency employs the temperature sampling technique to draw samples from the LLM. This sampling step can introduce noise, leading to predictions that are inconsistent with their corresponding explanations (Ye & Durrett, 2022).

---

[2]Future work would be suited to consider extending our work to generative tasks.

The identified limitations necessitate the need for new techniques to better harvest intermediate explanations for obtaining the final prediction. Towards this goal, we propose our framework EASE, which is tailored to tackle the aforementioned challenges. EASE is comprised with two techniques, explanation-aware ensemble and soft probability aggregation, to optimize the LLM's prediction accuracy when deriving final outcomes from multiple candidate explanations.

## 3.3 EXPLANATION-GUIDED ENSEMBLE

LLMs typically produce multiple explanations along with their predictions through a sampling process. Due to the intrinsic randomness of this sampling, the quality of these predictions can fluctuate. To address the potential pitfalls where erroneous explanations results in inaccurate predictions, we introduce the *explanation-aware ensemble* technique. This method estimates the significance of each class prediction based on its corresponding explanation. Consequently, our explanation-aware ensemble technique ensures that predictions linked with better explanations carry greater weight during the final prediction aggregation phase.

**LLM as Explanation Scorer** To evaluate various explanations, past research has either measured the lexical overlap between the explanation and the input text (Ye & Durrett, 2022) or employed models fine-tuned for NLI tasks (Chen et al., 2021; Li et al., 2023b). In contrast to these methods, which either overlook the deep semantics of explanations or require extra human-annotated data, our explanation scorer is developed based on the powerful LLM $\mathcal{M}$, directly harnessing its inherent linguistic and reasoning capabilities.

Given the original task input $x$ and one explanation $e$, we use the verbalizer $v_{\mathrm{pos}}(v_{\mathrm{neg}})$ to represent the class of this explanation being "*positive*" ("*negative*"). A "*positive*" explanation means this explanation can help the model reach correct answer and a "*negative*" explanation means the other way around. Then, we craft a supplementary prompt $\mathcal{T}_{\mathrm{score}} = $ "*Can this explanation be used to help the model answer the question?*" for LLM prompting. With the verbalizers and prompts, we effectively recast the problem of explanation scoring into determining the conditional probability of producing the verbalizer aligned with the positive label $v_{\mathrm{pos}}$, expressed as

$$\omega_e = p_\theta \left( y = v_{\mathrm{pos}} \mid \mathcal{T}_{\mathrm{score}}, x, e \right). \tag{3}$$

In this way, the score $\omega_e$ is normalized between 0 and 1 and a higher score indicates the explanation with better quality.

**Bootstrapped LLM Scorer** Although the above approach can already produce scores for each prediction, the score generated with the LLM $\mathcal{M}$ can still be biased and less precise (Wang et al., 2023b), especially under the zero-shot scenario where no demonstrations are provided. To mitigate the bias and generate reliable scores, we propose to provide additional examples to serve as "*positive*" and "*negative*" explanations to facilitate LLM scoring using the original few-shot demonstrations in $\mathcal{D}$.

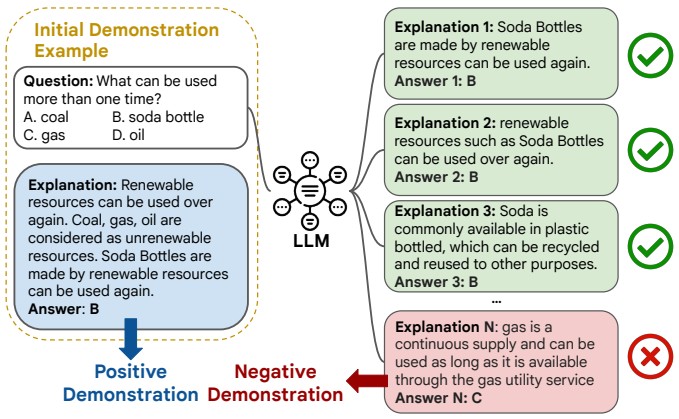

Figure 2: Bootstrapped LLM Scorer.

For each original demonstration instance, it is straightforward to obtain "*positive*" examples from the ground-truth explanation. Obtaining "*negative*" examples, on the other hand, can be more challenging as they are not explicitly provided. To tackle this issue, we exploit the assumption based on the utility of explanations: an ideal explanation should guide the model towards the accurate prediction of ground-truth labels (Wiegreffe et al., 2021). Consequently, it's reasonable to classify explanations leading to erroneous predictions as "negative". In practice, for every instance $(x_i, y_i) \in \mathcal{D}$, we randomly select $k$ (8 in this work) exemplars from the training set and then use these as demonstrations and generate a set of candidate pairs $\mathcal{C}_i = \{(e_{ij}, p_{ij})\}_{j=1}^{N}$ via sampling from the LLM. Then, if the explanation-

prediction pair $(e_{ij}, p_{ij})$ from $\mathcal{C}_i$ satisfies $y_i \neq p_{ij}$, we select $e_{ij}$ to serve as the negative explanation set $\mathcal{N}_i$ for $x_i$ as

$$\mathcal{N}_i = \{(e_{ij}, p_{ij}) \in \mathcal{C}_i \mid y_i \neq p_{ij}\}. \tag{4}$$

To finalize the demonstration set for the LLM scoring step, we balance between "*positive*" and "*negative*" explanations: only instances possessing negative explanations (i.e. with non-empty $\mathcal{N}_i$) are incorporated into the demonstrations; For every instance, a single negative explanation is chosen at random from the respective candidate set. This methodology produces a balanced demonstration set for LLM-based explanation scoring without requiring extra human annotations.

### 3.4 SOFT PROBABILITY AGGREGATION

In the preceding step, the primary objective is to assign a score to each prediction based on its associated explanation through the LLM $\mathcal{M}$. This process, however, does not account for directly modeling the LLM's output predictions. To bridge this gap, we propose *soft probability aggregation*, a simple and intuitive approach to resolve the discrepancy between the explanations and predictions — rather than aggregating over the raw predictions, it directly computes the sum of the probabilities associated with each potential label, expressed as

$$\widetilde{y} = \underset{y}{\arg\max} \ \sum_{j=1}^{N} p_\theta \left( y \mid \mathcal{P}, x, e_j \right). \tag{5}$$

The *soft probability aggregation* addresses the noise inherited in different LLM sampling-based decoding algorithms, resulting in a more accurate and refined final prediction.

### 3.5 SUMMARY

By plugging these two techniques together, we obtain the final prediction $\widetilde{y}$ for the test instance $x$ as

$$\widetilde{y} = \underset{y}{\arg\max} \ \sum_{j=1}^{N} \omega_{e_j} \times p_\theta \left( y \mid \mathcal{P}, x, e_j \right), \tag{6}$$

where $e_j$ is the intermediate explanations generated via Eq. 1, the $\omega_{e_j}$ is the weight for $e_j$ using the bootstrapped LLM scorer using Eq. 3, and $p_\theta \left( y \mid \mathcal{P}, x, e_j \right)$ is the soft probability generated using Eq. 5. Overall, calculating the score for each explanation and the soft probability both take an additional $O(N)$ time complexity. Fortunately, these two steps do not require additional model training and can be efficiently supported with distributed inference techniques in practice. Other than these two techniques, our framework keeps other components intact and can be plugged into most LLM backbones for empowering its in-context learning ability.

## 4 EXPERIMENTS

### 4.1 EXPERIMENT SETUPS

**Tasks** We evaluate our EASE framework on two types of tasks: natural language inference and question answering. Specifically, we use the following datasets: (1) **E-SNLI** (Camburu et al., 2018) is an enriched version of the Stanford Natural Language Inference (SNLI) corpus (Bowman et al., 2015), augmented with human-annotated natural language explanations for entailment relations; (2) **ANLI-R1/R2/R3** (Nie et al., 2020) is a set of three collections of adversarially generated NLI examples curated through a human-in-the-loop process; (3) **ECQA** (Aggarwal et al., 2021) is built upon CommonsenseQA benchmark (Talmor et al., 2019) and contains additional human-annotated question explanations; (4) **OpenbookQA** (Mihaylov et al. (2018)) is a QA dataset that requires comprehensive understanding and reasoning from open-book sources. As no ground-truth explanations are given, we use the provided facts for each question as the proxy explanations. (5) **StrategyQA** (Geva et al., 2021) focuses on reasoning over complex, multi-hop questions that often require strategic planning and decision-making.

**Baselines** We consider the following baselines: (1) **Standard In-context Learning (ICL)** (Brown et al., 2020): it solely uses the input-label pairs for few-shot learning without using natural language explanations. (2) **Predict-then-Explain (PE)** (Lampinen et al., 2022): it provides the explanation

Table 1: The main experiments results, where "BLS" stands for bootstrapped LLM scorer and "SPA" stands for soft probability aggregation. All results have passed the statistically significant test ($p < 0.05$) over baselines.

| Backbone | Methods | E-SNLI | ANLI-R1 | ANLI-R2 | ANLI-R3 | ECQA | StrategyQA | OpenbookQA | Average |
|---|---|---|---|---|---|---|---|---|---|
| PaLM 2-S | ICL (Brown et al., 2020) | 59.88 | 54.38 | 48.10 | 52.66 | 59.84 | 66.69 | 80.21 | 60.25 |
| | PE (Lampinen et al., 2022) | 71.02 | 62.59 | 55.18 | 57.17 | 74.39 | 71.75 | 79.70 | 67.40 |
| | EP (Wei et al., 2022b) | 64.53 | 57.40 | 53.00 | 53.33 | 72.11 | 72.40 | 81.38 | 64.88 |
| | Self-consistency (Wang et al., 2023c) | 68.68 | 65.40 | 56.49 | 59.00 | 74.48 | 76.94 | 83.47 | 69.21 |
| | FLamE (Zhou et al., 2023) | 67.58 | 60.36 | 52.00 | 50.15 | 72.80 | 75.33 | 80.14 | 65.48 |
| | EASE | **75.01** | 66.48 | **59.66** | **64.33** | **75.59** | 78.23 | **84.10** | **71.92 (↑3.91%)** |
| | EASE w/o BLS | 73.84 | **66.84** | 58.74 | 62.66 | 75.17 | **78.40** | 83.91 | 71.37 |
| | EASE w/o SPA | 69.82 | 66.77 | 58.50 | 62.50 | 75.42 | 78.33 | 83.68 | 70.73 |
| PaLM 2-L | ICL (Brown et al., 2020) | 87.42 | 79.00 | 68.33 | 65.65 | 81.29 | 81.13 | 91.17 | 79.14 |
| | PE (Lampinen et al., 2022) | 88.84 | 80.55 | 71.49 | 68.33 | 83.13 | 83.19 | 92.46 | 81.14 |
| | EP (Wei et al., 2022b) | 84.59 | 79.03 | 67.99 | 67.66 | 80.51 | 85.45 | 89.74 | 79.28 |
| | Self-consistency (Wang et al., 2023c) | 87.34 | 81.29 | 73.16 | 70.16 | 82.67 | 87.85 | 92.88 | 82.19 |
| | FLamE (Zhou et al., 2023) | 83.23 | 71.85 | 58.50 | 56.83 | 80.26 | 84.79 | 93.14 | 75.51 |
| | EASE | **89.42** | **83.69** | **76.16** | **74.00** | **83.65** | **89.90** | **93.93** | **84.40 (↑2.69%)** |
| | EASE w/o BLS | 88.94 | 82.87 | 75.60 | 72.66 | 83.42 | 89.34 | 93.72 | 83.79 |
| | EASE w/o SPA | 88.21 | 82.59 | 73.83 | 71.33 | 83.42 | 89.35 | 93.51 | 83.18 |

after the labels for each instance when constructing prompts for demonstrations. During the inference stage, it generates the explanation after the prediction. (3) **Explain-then-Predict (EP)** (Wei et al., 2022b): it is the standard chain-of-thought pipeline which provides an explanation before the label for demonstrations. During the inference stage, it first generates an explanation, then followed by the prediction. Note that for both PE and EP method, we use greedy sampling to obtain the explanation and prediction. (4) **Self-consistency** (Wang et al., 2022; 2023c): it improves over the standard EP pipeline by aggregating over multiple explanations from LLMs to enhance the robustness of the results. (5) **FLamE** (Zhou et al., 2023) is a recent LLM few-shot learning method that generates multiple label-conditioned explanations and determines the final prediction based on the label that achieves the highest logit after reviewing all explanations for the given instance[3].

**Implementation Details** In our main experiments, we use PaLM2-S and PaLM2-L (Anil et al., 2023) as the backbone model. Results on more (open source) backbone models are reported in Section 4.3. For each dataset, we set the size of few-shot examples to 48 following (Zhou et al., 2023; Marasovic et al., 2022), and fit as many instances as possible during inference until reached the maximum length. As the LLM is often sensitive to the selection of few-shot examples (Ye & Durrett, 2023; Liu et al., 2022), for each dataset, we create 5 splits from the original dataset, each containing 300 test examples, and report the average performance over 5 splits. During sampling, we set the default temperate to $t = 0.7$ and sample $N = 9$ candidate explanations for each instance.

## 4.2 OVERALL RESULTS

Table 1 exhibits the performance of EASE and baselines on seven datasets using PaLM 2-S and PaLM 2-L as the backbone. From the results, we have the following findings: **First**, we can see that leveraging explanations often improves LLM in-context learning. This enhancement is particularly pronounced when the final prediction is aggregated from multiple predictions sampled from the LLM[4]. Conversely, the standard EP pipeline sometimes even hurts the performance, especially for larger models. **Second**, despite its complex design, the latest baseline FLamE often falls short compared to other baselines, which suggests that fine-tuning an additional classifier is particularly important for FLamE and it might be less compatible with the LLM in-context learning framework. **Third**, we notice that EASE can consistently outperform all other methods across both the PaLM 2-S and PaLM 2-L backbones in nearly all datasets, which indicates that EASE provides a reliable way to improve LLM in-context learning over different tasks. **Finally**, When comparing EASE with its own variants (e.g. w/o BLS and SPA), it's observed that the original EASE consistently holds an advantage, indicating the necessity of both PW and SA components in maximizing performance.

## 4.3 RESULTS ON OPEN-SOURCE MODELS

In order to demonstrate the generalizability of our EASE framework, as well as promote reproducibility, we extend our investigations to open-source LLMs including FLAN-UL2 (Tay et al.,

---

[3]In the original FLamE paper, the RoBERTa is used for final classification. For a fair comparison, we adjusted FLamE to use the in-context LLM as the classifier.

[4]We have also tried to incorporate multiple explanations with temperature sampling, but we find performance drops. This is because the prediction of the PE pipeline can not benefit from generated explanations.

Table 2: The main experiments results on open-source models, where "BLS" stands for bootstrapped LLM scorer and "SPA" stands for soft probability aggregation. All results have passed the statistically significant test ($p < 0.05$) over baselines.

| Model ($\rightarrow$) | FLAN-UL2 (20B) | Llama-2 (7B) | | | | | | | |
|---|---|---|---|---|---|---|---|---|---|
| Dataset ($\rightarrow$) | StrategyQA | E-SNLI | ANLI-R1 | ANLI-R2 | ANLI-R3 | ECQA | StrategyQA | OpenbookQA | Avg. |
| ICL (Brown et al., 2020) | 61.76 | 51.14 | 34.58 | 36.05 | 27.48 | 45.48 | 53.81 | 47.48 | 42.29 |
| PE (Lampinen et al., 2022) | 73.42 | 54.25 | 37.83 | 37.50 | 34.37 | 52.33 | 56.21 | 56.48 | 47.00 |
| EP (Wei et al., 2022b) | 75.46 | 56.90 | 35.41 | 39.16 | 36.04 | 54.45 | 57.17 | 44.35 | 46.21 |
| Self-consistency (Wang et al., 2023c) | 76.01 | 58.79 | 40.16 | 40.16 | 36.16 | 55.14 | 57.12 | 60.87 | 49.77 |
| FLamE (Zhou et al., 2023) | 72.17 | 49.32 | 36.83 | 35.16 | 36.50 | 45.11 | 57.70 | 46.23 | 43.84 |
| EASE | **78.70** (↑ 3.55%) | 60.80 | **44.50** | **41.66** | **41.33** | 60.45 | 59.81 | 64.43 | **53.28** (↑ 7.05%) |
| EASE w/o BLS | 77.31 | 59.54 | 43.45 | 41.33 | 40.33 | 60.34 | 59.62 | **65.06** | 52.81 |
| EASE w/o SPA | 77.78 | 58.50 | 41.33 | 40.16 | 35.33 | 54.97 | 57.40 | 61.71 | 49.91 |

Table 3: The study on different scoring approaches. Note that to ensure fair comparison, we do not use soft probability aggregation for our method and baselines.

| Dataset ($\rightarrow$) | E-SNLI | | OpenbookQA | | StrategyQA |
|---|---|---|---|---|---|
| Model ($\rightarrow$) | PaLM 2-S | PaLM 2-L | PaLM 2-S | PaLM 2-L | FLAN-UL2 |
| EASE | **69.82** | **83.68** | **83.68** | 93.51 | **78.70** |
| EASE w/ PE Negative | 68.90 | 83.91 | 83.54 | **93.93** | 78.06 |
| LLM Zero-shot Scoring (Fu et al., 2023a) | 66.84 | 81.77 | 81.38 | 88.50 | 75.15 |
| LLM Pairwise Scoring (Qin et al., 2023) | 69.25 | 82.97 | 82.97 | 93.14 | 76.93 |
| Lexical Scoring (Ye & Durrett, 2022) | 67.72 | 83.54 | 82.66 | 93.72 | 75.34 |
| NLI Scoring (Chen et al., 2021) | 64.87 | 81.89 | 82.21 | 91.52 | 76.11 |

2023)[5] and Llama-2-7b (Touvron et al., 2023). Both models have publicly accessible weights[6]. As exhibited in Table 2, we observe that these two models generally perform worse than the PaLM 2 model in the main experiments, as they have fewer parameters, and thus may not perform well on these challenging NLU benchmarks. Despite this, the experiment results still align with our prior findings, demonstrating that our proposed techniques can consistently yield performance enhancements across these open-source LLMs.

## 4.4 STUDY ON EXPLANATION-AWARE ENSEMBLE

We perform additional experiments to further understand the benefit of the explanation-aware ensemble, and the result is shown in Table 3.

**Performance w/ Different Scoring Methods** We first compare our LLM-based explanation scorer with a few alternative methods including (1) *lexical scoring*, which estimates the reliability of explanations via the lexical gap (Ye & Durrett, 2022), and (2) *NLI Scoring* that uses an NLI model to verify the reliability of explanations. In this work, we use MT5-XXL (Xue et al., 2021) fine-tuned on NLI datasets as the scorer. Overall, we observe that our model outperforms these models in most of the cases, indicating that LLM has a strong capacity for estimating the quality of the explanations. In addition, we observe that pairwise scoring does not perform well for weighting the predictions. This is because it was originally proposed for text ranking tasks, while there are many differences between it and our scenarios, including input formats and relevance signals.

**Performance w/ Different Bootstrapping Strategies** To justify the design of leveraging the Explain-then-Predict (EP) pipeline to generate negative demonstrations, we also consider other ways including removing demonstrations as well as using the Predict-then-Explain (PE) pipeline. Overall, in many cases, using the EP pipeline leads to better results, as we observe that the PE pipeline sometimes causes the *false negative* issue: it will first generate incorrect predictions but followed with reasonable explanations. However, when the model performs reasonably well (e.g. PaLM 2-L on OpenbookQA), then it may make less erroneous prediction during the bootstrapping step, which may lead to insufficient training signals for EASE to perform well. In addition, no matter whether PE and EP is used, they both largely outperform the baseline where no demonstration is given, necessitating the role of demonstration for explanation-aware ensembling.

**Score Distribution of Explanations** To delve deeper into the scores assigned to each explanation and justify that better scores are assigned to explanations with correct answers, we plot the score

---

[5]Link: https://github.com/google-research/google-research/tree/master/ul2. We only test on StrategyQA dataset since FLAN-UL2 has been fine-tuned on labeled data from other datasets, thus violating the few-shot setting.

[6]Link: https://huggingface.co/meta-llama/Llama-2-7b.

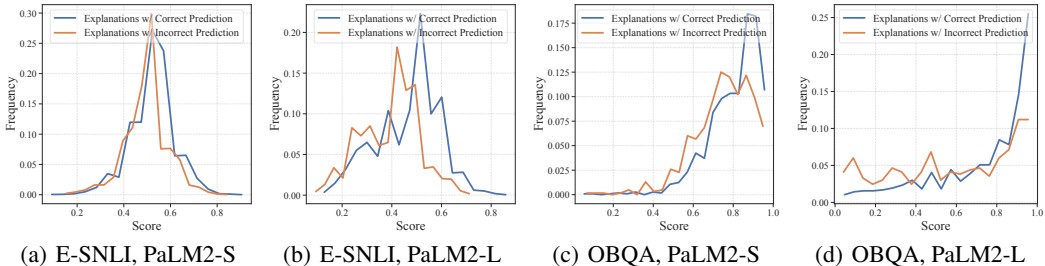

(a) E-SNLI, PaLM2-S    (b) E-SNLI, PaLM2-L    (c) OBQA, PaLM2-S    (d) OBQA, PaLM2-L

Figure 3: The score distribution for bootstrapped LLM scorer. OBQA stands for OpenbookQA.

distribution for explanations with correct predictions[7] in Figure 3. Overall, we observe that explanations that lead to correct answers generally have higher scores — the score distribution is more skewed towards higher values. Besides, the score distribution using PaLM2-L on explanation with correct and incorrect predictions are more separable, indicating larger models tend to have better scoring performance.

**Human Study on Explanations** We conduct additional human studies to further investigate whether the scores generated by LLM are aligned with human preferences. For each instance, we sample two explanations with *different* predictions as $\{(e_1, p_1), (e_2, p_2)\}$, with one being correct. We compare our approach and two baselines (NLI model, lexical overlap) with human raters: for each pair of explanations, we first ask four humans to determine which explanation is better and use $c_i$ ($i = 1, 2$) to denote the number of raters that select $e_i$ as the better one. Then, we use different models to estimate the score for explanations separately, denoted as $(s_{e_1}, s_{e_2})$. The final judge of "Win-Tie-Lose" is determined to be:

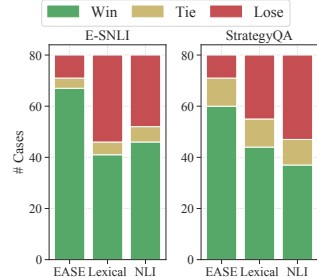

Figure 4: Human Evaluation.

$$r = \begin{cases} \text{win,} & \text{if } (c_1 > c_2 \text{ and } s_{e_1} > s_{e_2}) \text{ or } (c_1 < c_2 \text{ and } s_{e_1} < s_{e_2}); \\ \text{tie,} & c_1 = c_2; \\ \text{lose,} & \text{if } (c_1 < c_2 \text{ and } s_{e_1} > s_{e_2}) \text{ or } (c_1 > c_2 \text{ and } s_{e_1} < s_{e_2}). \end{cases} \quad (7)$$

On two datasets, we randomly select 80 instances, and the final results are shown in Figure 4. The cohen's kappa among human raters are 0.75 (E-SNLI) and 0.64 (StrategyQA), which stands for "*substantial agreement*". Overall, we observe that EASE aligns with human preferences the best, indicating its better ability to be the proxy for explanation quality estimation. We display more examples on generated explanations and the scores in Appendix F.1.

## 4.5 STUDY ON SOFT PROBABILITY AGGREGATION

The premise behind soft probability aggregation is the potential inaccuracy in the prediction token due to temperature sampling variability. To verify this, we calculate the proportion of cases where the prediction token $p_i$ is different than the prediction $p_i \neq \arg\max p(\cdot | \mathcal{P}, x, e_i)$. Overall, as exhibited in Table 4, we observe that such inconsistency predic-

Table 4: The study on different probability aggregation approaches. Note that we do not use explanation-aware ensemble for our method and baselines.

| Dataset ($\rightarrow$) | E-SNLI | | OpenbookQA | | StrategyQA |
|---|---|---|---|---|---|
| Model ($\rightarrow$) | PaLM 2-S | PaLM 2-L | PaLM 2-S | PaLM 2-L | FLAN-UL2 |
| Inconsistency Ratio | 14.60% | 10.06% | 13.96% | 10.71% | 10.00% |
| EASE | 73.84 | 88.21 | 83.91 | 93.72 | 78.70 |
| w/ argmax | 73.20 | 87.90 | 83.68 | 93.51 | 78.42 |
| Cond. Gen (Li et al., 2023a) | 70.77 | 82.20 | 78.07 | 84.38 | 72.80 |

tions appear in 10% to 15% of the cases, which is not rare in practice. By using the soft score, we observe that it will consistently lead to performance boosts. The gain is more evident when the inconsistency issue is more severe — on E-SNLI dataset using PaLM 2-S as the backbone, there exist around 15% examples with inconsistent predictions. When incorporating soft probability aggregation, we observe a notable performance gain (from 68.68% to 73.84%). When compared to other

---

[7]To eliminate the effect of the sampling randomness, we calculate the prediction based on the soft probability using Eq. 5.

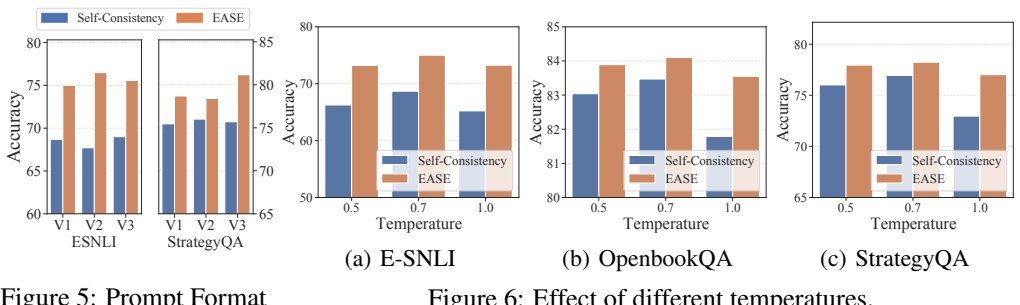

Figure 5: Prompt Format

(a) E-SNLI        (b) OpenbookQA        (c) StrategyQA

Figure 6: Effect of different temperatures.

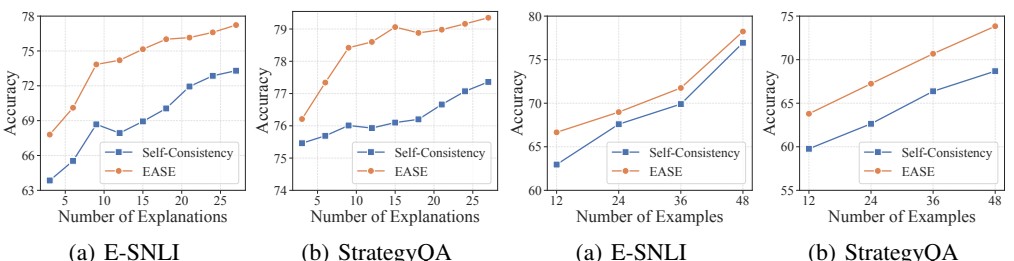

(a) E-SNLI        (b) StrategyQA        (a) E-SNLI        (b) StrategyQA

Figure 7: Effect of number of explanations.        Figure 8: Effect of number of demonstrations.

methods for prediction correction, such as using the hard prediction (*i.e.* $\arg\max p(\cdot|\mathcal{P}, x, e_i)$) or generation probability conditioned on different verbalizers, EASE also achieves better performance. More case studies on using soft probabilities are deferred to Appendix F.2.

### 4.6 ADDITIONAL STUDIES

As EASE relies on several key components such as prompts and sampling steps, in this section, we study their effect on the final prediction performance, using PaLM 2-S as the backbone model.

**Effect of the Sampling Temperatures and Prompt Templates** We study the robustness EASE to different prompt templates by choosing three different prompt formats from (Bach et al., 2022) (the details are in Appendix C.3) on two datasets. Overall, from Figure 5 we observe that EASE is robust to them as all of the prompt formats lead to performance gains when compared to the strongest baseline self-consistency. Similarly, in Figure 6, we observe that EASE also performs better than baseline under all temperature settings, further justify its robustness across different settings.

**Effect of the Number of Generated Explanations $N$** In Figure 7, we examine the influence of the number of explanations. On both datasets, increasing the explanations generally improves the performance, while EASE achieves better performance than the baselines using only 30% - 40% of the generated explanations, which can reduce the burden of sampling massive explanations while maintaining the performance.

**Effect of the Size of demonstrations $K$** Figure 8 illustrates the performance with different size of demonstrations. By increasing the number of demonstration $K$, the performance gradually increases, while EASE achieves performance gains under all value of $K$.

## 5 CONCLUSION AND DISCUSSION

In this work, we empower LLM's in-context learning ability with natural language explanations. Specifically, we design explanation-aware ensemble to weight multiple predictions using their associated explanations and realize this idea using a bootstrapped LLM scorer. In addition, we leverage a soft probability aggregation scheme to mitigate the issue of inconsistent predictions for ensembling. We conduct extensive experiments on seven datasets from a diverse task set and show our proposed framework can outperform previous state-of-the-art methods using four LLMs as backbones.

Notably, while EASE augments in-context learning by weighting predictions through explanations, it does not refine the explanation's content. For future works, it is potential to leverage self-refinement (Madaan et al., 2023), debating (Du et al., 2023), or negotiation (Fu et al., 2023b) to elevate explanation quality and strengthen the model's reasoning abilities.

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

# A    LIMITATIONS

In this work, our primary goal is to identify the existing issues to better leverage explanations to empower in-context learning. While our approach has shown promise, it also comes with increased computational demands, as both explanation-aware ensemble and soft probability aggregation steps require additional computation overhead. Future work could explore designing more powerful prompts to let LLMs directly output the suffix tokens as quality score (Tian et al., 2023). Additionally, our methodology depends on the logits returned in both the explanation-aware ensemble and soft probability aggregation processes, making it less suitable to directly adapt to black-box LLMs (e.g. ChatGPT, OpenAI (2023)). To approximate the soft score, one strategy is to set the temperature to a non-zero value and conduct multiple sampling steps, then use the frequency of the corresponding verbalizers as the proxy of the score.

Besides, the key assumption of EASE is that different explanations are of diverse quality, while those explanation leads to correct predictions tend to be of higher quality. We mainly conduct empirical experiments to support this point, yet there often exists multiple facets to evaluate the quality of free-text explantions (Chen et al., 2023a;b; Sun et al., 2022). More in-depth metrics are needed to faithfully evaluate the quality of free-text explanations and reveal the true inner workings of EASE.

Additionally, as the few-shot in-context learning performance is often sensitive to the selection of the demonstrations, EASE may not fully outperform its variants under all data splits. Another promising research direction is to automatically construct few-shot demonstrations to further improve the performance of EASE.

# B    DATASETS DETAILS

The seven benchmarks in our experiments are all publicly available. Below are the links to downloadable versions of these datasets.

- **E-SNLI**: https://huggingface.co/datasets/esnli;
- **ANLI R1/R2/R3**: https://github.com/facebookresearch/anli;
- **ECQA**: https://github.com/allenai/feb;
- **OpenbookQA**: https://huggingface.co/datasets/openbookqa;
- **StrategyQA**: for StrategyQA we use the question-only set from the link https://github.com/google/BIG-bench/blob/main/bigbench/benchmark_tasks/strategyqa

By default, we sample few-shot demonstrations from the train set and sample from the test split for all datasets. For OpenbookQA, as the original dataset only contains 500 test examples, in each split we use 100 examples. For ANLI, as some of the examples contain no explanations, while the explanations for some examples include task-irrelevant information such as 'I think the computer was confused because so many of the words were similar to the description'. To reduce the effect of such examples, we remove those examples occurs with term 'the system', 'the computer', 'the model', 'the AI', and manually checked all the few-shot demonstrations to ensure that there is no such information in explanations.

# C    PROMPT FORMATS

In this section, we list the prompts used in our experiments.

## C.1    PROMPT FORMAT FOR IN-CONTEXT LEARNING

In this step, we list the prompt for generating the explanations and predictions. Many of the prompt formats are adapted from (Bach et al., 2022). Note that the blue text is instance-dependent, while the red text is the model's expected output.

### C.1.1 E-SNLI

Listing 1: Prompt Format for E-SNLI dataset, standard in-context learning.

```
In this task, given a premise and a hypothesis, your job is to
determine whether the hypothesis can be inferred from the premise.

# demonstrations (no more than 48)
Based on the premise: [premise], can we infer the hypothesis:
 [hypothesis] from the premise? Choose among Yes, Maybe, and No.
Answer: [Answer]

# test examples
Based on the premise: [premise], can we infer the hypothesis:
 [hypothesis] from the premise? Choose among Yes, Maybe, and No.
Answer: [Answer]
```

Listing 2: Prompt Format for E-SNLI dataset, using predict-then-explain pipeline.

```
In this task, given a premise and a hypothesis, your job is to
determine whether the hypothesis can be inferred from the premise.

# demonstrations (no more than 48)
Based on the premise: [premise], can we infer the hypothesis:
 [hypothesis] from the premise? Choose among Yes, Maybe, and No.
Answer: [Answer]
Explanation: [Explanation]

# test examples
Based on the premise: [premise], can we infer the hypothesis:
 [hypothesis] from the premise? Choose among Yes, Maybe, and No.
Answer: [Answer]
Explanation:  [Explanation]
```

Listing 3: Prompt Format for E-SNLI dataset, using explain-then-predict pipeline.

```
In this task, given a premise and a hypothesis, your job is to
determine whether the hypothesis can be inferred from the premise.

# demonstrations (no more than 48)
Based on the premise: [premise], can we infer the hypothesis:
 [hypothesis] from the premise? Choose among Yes, Maybe, and No.
Answer: [Answer]
Explanation: [Explanation]

# test examples
Based on the premise: [premise], can we infer the hypothesis:
 [hypothesis] from the premise? Choose among Yes, Maybe, and No.
Explanation: [Explanation]
Answer:  [Answer]
```

## C.1.2  ANLI

Listing 4: Prompt Format for ANLI dataset, standard in-context learning.

```
In this task, given a premise and a hypothesis, your job is to
determine whether the hypothesis can be inferred from the premise.

# demonstrations (no more than 48)
Based on the premise: [premise], can we infer the hypothesis:
 [premise] from the premise? Choose among Yes, Maybe, and No.
Answer: [Answer]

# test examples
Based on the premise: [premise], can we infer the hypothesis:
 [premise] from the premise? Choose among Yes, Maybe, and No.
Answer: [Answer]
```

Listing 5: Prompt Format for ANLI dataset, using predict-then-explain pipeline.

```
In this task, given a premise and a hypothesis, your job is to
determine whether the hypothesis can be inferred from the premise.

# demonstrations (no more than 48)
[premise], Based on the previous passage, is it true that
[hypothesis]? Choose among Yes, Maybe, and No.
Answer: [Answer]
Explanation: [Explanation]

# test examples
[premise], Based on the previous passage, is it true that
[hypothesis]? Choose among Yes, Maybe, and No.
Answer: [Answer]
Explanation:  [Explanation]
```

Listing 6: Prompt Format for ANLI dataset, using explain-then-predict pipeline.

```
In this task, given a premise and a hypothesis, your job is to
determine whether the hypothesis can be inferred from the premise.

# demonstrations (no more than 48)
[premise], Based on the previous passage, is it true that
[hypothesis]? Choose among Yes, Maybe, and No.
Answer: [Answer]
Explanation: [Explanation]

# test examples
[premise], Based on the previous passage, is it true that
[hypothesis]? Choose among Yes, Maybe, and No.
Explanation: [Explanation]
Answer:  [Answer]
```

### C.1.3   ECQA & OPENBOOKQA

As both ECQA & OpenbookQA are multi-choice classification tasks, we use the same prompt formats for them.

Listing 7: Prompt format for multi-choice QA, standard in-context learning.

```
In this task, your job is to first read the question as well as
the candidate choices. Then, choose one answer from the choices
for the question.

# demonstrations (no more than 48)
Given the following options, what do you think is the correct
answer to the question below?
Question:  [question]
Choices: [choices]
Answer: [Answer]

# test examples
Given the following options, what do you think is the correct
answer to the question below?
Question:  [question]
Choices: [choices]
Answer: [Answer]
```

Listing 8: Prompt format for multi-choice QA, using predict-then-explain pipeline.

```
In this task, your job is to first read the question as well as
the candidate choices. Then, choose one answer from the choices
for the question.

# demonstrations (no more than 48)
Given the following options, what do you think is the correct
answer to the question below?
Question:  [question]
Choices: [choices]
Answer: [Answer]
Explanation: [Explanation]

# test examples
Given the following options, what do you think is the correct
answer to the question below?
Question:  [question]
Choices: [choices]
Answer: [Answer]
Explanation:  [Explanation]
```

Listing 9: Prompt format for multi-choice QA, using explain-then-predict pipeline.

```
In this task, your job is to first read the question as well as
the candidate choices. Then, choose one answer from the choices
for the question.

# demonstrations (no more than 48)
Given the following options, what do you think is the correct
answer to the question below?
Question:  [question]
Choices: [choices]
Explanation: [Explanation]
Answer: [Answer]

# test examples
Given the following options, what do you think is the correct
answer to the question below?
Question:  [question]
Choices: [choices]
Explanation: [Explanation]
Answer:  [Answer]
```

### C.1.4 STRATEGYQA

Listing 10: Prompt format for StrategyQA, standard in-context learning.

```
In this task, given a question, you need to answer True or False.
# demonstrations (no more than 48)
For the question: '[question]', do you think it is the True or
False?
Answer: [Answer]

# test examples
For the question: '[question]', do you think it is the True or
False?
Answer: [Answer]
```

Listing 11: Prompt format for StrategyQA, using predict-then-explain pipeline.

```
In this task, given a question, you need to answer True or False.
# demonstrations (no more than 48)
For the question: '[question]', do you think it is the True or
False?
Answer: [Answer]
Explanation: [Explanation]

# test examples
For the question: '[question]', do you think it is the True or
False?
Answer: [Answer]
Explanation:  [Explanation]
```

Listing 12: Prompt format for StrategyQA, using explain-then-predict pipeline.

```
In this task, given a question, you need to answer True or False.

# demonstrations (no more than 48)
For the question: '[question]', do you think it is the True or
False?
Explanation: [Explanation]
Answer: [Answer]

# test examples
For the question: '[question]', do you think it is the True or
False?
Explanation: [Explanation]
Answer:  [Answer]
```

## C.2 PROMPT FORMAT FOR EXPLANATION-AWARE ENSEMBLE.

Listing 13: Prompt format for LLM Scoring. Note that we use the probability of the 'Answer' token as the proxy for the quality score.

```
In this task, you will be given the input for the [task_name] task
, your job is to determine whether the explanation provided is a
good one for the given input. Please consider the explanation's
coherence, informativeness, and consistency with the prediction to
 evaluate its quality.

# demonstrations (no more than 48)
For '[task input]', can you determine whether the explanation is a
 good one for the given [task]?
Explanation: [Explanation]
Answer: [Answer] [Yes or No]

# test examples
For '[task input]', can you determine whether the explanation is a
 good one for the given [task]?
Explanation: [Explanation]
Answer: [Answer]
```

## C.3 ADDITIONAL PROMPT FORMAT USED IN PROMPT SENSITIVITY STUDY

In section 4.6, we have studied the effect of different prompt templates. Here we list them in the following lists.

Listing 14: Prompt Format 2 for E-SNLI dataset

```
In this task, given a premise and a hypothesis, your job is to
determine whether the hypothesis can be inferred from the premise.

# demonstrations (no more than 48)
Based on [premise], does it follow that [hypothesis]? Choose among
 Yes, Maybe, and No.
Answer: [Answer]
Explanation: [Explanation]

# test examples
Based on [premise], does it follow that [hypothesis]? Choose among
 Yes, Maybe, and No.
Explanation: [Explanation]
Answer:  [Answer]
```

Listing 15: Prompt Format 3 for E-SNLI dataset

```
In this task, given a premise and a hypothesis, your job is to
determine whether the hypothesis can be inferred from the premise.

# demonstrations (no more than 48)
Based on the premise [premise], can we conclude the hypothesis
that [hypothesis]? Choose among Yes, Maybe, and No.
Answer: [Answer]
Explanation: [Explanation]

# test examples
Based on the premise [premise], can we conclude the hypothesis
that [hypothesis]? Choose among Yes, Maybe, and No.
Explanation: [Explanation]
Answer:  [Answer]
```

Listing 16: Prompt format 2 for StrategyQA, using explain-then-predict pipeline.

```
In this task, given a question, you need to answer True or False.

# demonstrations (no more than 48)
Answer the question: '[question]', by True or False.
Explanation: [Explanation]
Answer: [Answer]

# test examples
Answer the question: '[question]', by True or False.
Explanation: [Explanation]
Answer:  [Answer]
```

Listing 17: Prompt format 3 for StrategyQA, using explain-then-predict pipeline.

```
In this task, given a question, you need to answer True or False.

# demonstrations (no more than 48)
EXAM: Answer by True of False.
Question: '[question]'
Explanation: [Explanation]
Answer: [Answer]

# test examples
EXAM: Answer by True of False.
Question: '[question]'
Explanation: [Explanation]
Answer:  [Answer]
```

# D   HUMAN EVALUATION

Here we provide the guidelines for human evaluation

Listing 18: Human Evaluation Guideline for E-SNLI dataset.

```
For this explanation grading task, given the task input (e.g. the
premise and hypothesis for the NLI task and the question for the
QA task), ground-truth answer, as well as a pair of explanations
from the LLM, you job is to determine which explantion will reach
the ground-truth answer for that input.
For the E-SNLI dataset, your task is to predict if the hypothesis
is entailed/neutral/contradicts the premise.
```

Listing 19: Human Evaluation Guideline for StrategyQA dataset.

```
For this explanation grading task, given the task input (e.g. the
premise and hypothesis for the NLI task and the question for the
QA task), ground-truth answer, as well as a pair of explanations
from the LLM, you job is to determine which explantion will reach
the ground-truth answer for that input.
For the strategyQA dataset, your task is to answer the question
with 'True' or 'False'.
```

# E   STUDIES ON VERBALIZERS FOR BOOTSTRAPPED LLM SCORER

We investigate the role of verbalizers for representing the "*positive*" and "*negative*" explanations. We consider three set of verbalizers, namely V1:"*Yes*" and "*No*", V2: "*True*" and "*False*", and V3: "*Foo*" and "*Jaa*" using symbolic tuning (Wei et al., 2023). Using PaLM 2-S as the backbone, we observe that the original "*Yes*" and "*No*" generally perform better. Symbolic tuning does not work as well as other verbalizers with concrete semantics, indicating it may not be strong enough for the explanation scoring task.

Table 5: Verbalizer Study for Bootstrapped LLM Scorer, using PaLM 2-S as the backbone.

| Template | V1 | V2 | V3 |
|---|---|---|---|
| E-SNLI | 75.01 | 73.75 | 74.12 |
| StrategyQA | 78.40 | 78.23 | 76.75 |

Table 6: Case study I for explanation-aware ensemble on E-SNLI Dataset.

| | |
|---|---|
| **Premise**: A man is working on a computer while two people sit and talk in front. **Hypothesis**: The two people sat and chatted for a bit while the IT guy removed the virus. **Ground-truth Label**: Neutral, **Majority Voting Prediction**: Entail. | |
| **Explanation 1**: The two people cannot be sitting and talking in front while the IT guy removes the virus. **Prediction 1**: Contradict | Score 1: 0.468 |
| **Explanation 2**: There is no proof that there is an IT Person removed the virus. **Prediction 2**: Neutral | Score 2: 0.562 |
| **Explanation 3**: The two people were sitting and talking. **Prediction 3**: Entail | Score 3: 0.369 |
| **Explanation 4**: There is no evidence that the man is an IT guy, or that he is removing a virus. **Prediction 4**: Neutral | Score 4: 0.677 |
| **Explanation 5**: The two people could be talking and the IT guy could be removing the virus. **Prediction 5**: Entail | Score 5: 0.488 |
| **Explanation 6**: But it's not given that the IT guy is removing a virus. **Prediction 6**: Neutral | Score 6: 0.612 |
| **Explanation 7**: The man on the computer is the IT guy. **Prediction 7**: Entail | Score 7: 0.468 |
| **Explanation 8**: The people sitting and talking are not related to the man working on the computer. **Prediction 8**: Contradict | Score 8: 0.447 |
| **Explanation 9**: The man sitting at the computer is an IT person and is removing the virus. **Prediction 9**: Entail | Score 9: 0.455 |

Table 7: Case study II for explanation-aware ensemble on StrategyQA Dataset.

| | |
|---|---|
| **Question**: Is Hermione Granger eligible for the Order of the British Empire? **Ground-truth Label**: False, **Majority Voting Prediction**: True. | |
| **Explanation 1**: The Order of the British Empire is a British order of chivalry rewarding contributions to the arts and sciences, work with charitable and welfare organizations and public service outside the civil service. Hermione Granger is a character in the Harry Potter series by J K. **Prediction**: True | Score 1: 0.426 |
| **Explanation 2**: The Order of the British Empire is a British Order of Chivalry. Hermione Granger is a fictional character living in a world of magic and wizards. The Order of the British Empire has never been awarded to a fictional character. **Prediction**: False | Score 2: 0.655 |
| **Explanation 3**: The Order of the British Empire was founded in 1917 by King George V. Hermione Granger is a British citizen To be eligible for the Order of the British Empire, a person must have contributed to the arts, science, medicine, charity, industry or government. **Prediction**: True | Score 3: 0.489 |
| **Explanation 4**: Hermione Granger is a fictional character from the Harry Potter books, not a British citizen. **Prediction**: False | Score 4: 0.678 |
| **Explanation 5**: Hermione Granger is a fictional character in the Harry Potter books. The Order of the British Empire is a British award that can only be received by a British subject . **Prediction**: False | Score 5: 0.650 |
| **Explanation 6**: Hermione Granger is a fictional character in the Harry Potter series Order of the British Empire honors people who have made a major contribution to the arts, science, medicine, charity, or industry. Hermione Granger has contributed to the arts by authoring a book on the history of magic. **Prediction**: True | Score 6: 0.406 |
| **Explanation 7**: Hermione Granger is a British citizen The Order of the British Empire is an order of chivalry. **Prediction**: True | Score 7: 0.452 |
| **Explanation 8**: The Order of the British Empire is a knighthood. Hermione Granger is a Hogwarts student. **Prediction**: True | Score 8: 0.406 |
| **Explanation 9**: Hermione Granger is a Hogwarts student and does not have British citizenship. **Prediction**: False | Score 9: 0.602 |

# F ADDITIONAL CASE STUDIES

## F.1 CASE STUDY ON EXPLANATION-AWARE ENSEMBLE

Table 6 and 7 give an example of Explanation-aware Ensemble process on E-SNLI dataset. Take the Table 6 as an example, where the original prediction using majority voting is "*Entailment*". By leveraging LLM to score each explanation, the LLM is able to reduce the effect of some unreliable explanations: for example, explanation 3 copies a part of the premise without providing additional reasoning, thus it receives a low score (0.369). On the contrary, explanation 4 correctly identifies the key reasoning step, thus receiving a higher score (0.677). After weighted majority voting, we are able to obtain the prediction "*Neutral*", which is the ground-truth answer.

## F.2 CASE STUDY ON SOFT PROBABILITY AGGREGATION

Table 8 and 9 provide examples of Soft Probability Aggregation applied to the ESNLI and Strate-gyQA Datasets, respectively. A notable observation is an occasional mismatch between predicted labels and their respective highest probability scores, indicating the potential for inconsistency in

Table 8: Case study for Soft Probability Aggregation on ESNLI Dataset. The mismatched prediction is highlighted in red.

| | |
|---|---|
| **Premise**: A Seattle firefighter standing in front of his truck.
**Hypothesis**: The man is standing in front of the ambulance.
**Ground-truth Label**: Contradict,  **Majority Voting Prediction**: Neutral. | |
| **Explanation 1**: The man is standing in front of the truck and the man is standing in front of the ambulance.
**Prediction**: Neutral | Score 1: [0.369, 0.419, 0.175] |
| **Explanation 2**: An ambulance is different than a truck.
**Prediction**: Contradict | Score 2: [0.001, 0.042, 0.953] |
| **Explanation 3**: The Seattle firefighter is standing in front of his truck. The truck is not necessarily an ambulance.
**Prediction**: Neutral | Score 3: [0.002, 0.880, 0.105] |
| **Explanation 4**: Ambulances are not fire trucks.
**Prediction**: Neutral | Score 4: [0.001, 0.372, 0.614] |
| **Explanation 5**: The guy is standing in front of a truck.
**Prediction**: Entail | Score 5: [0.391, 0.194, 0.378] |
| **Explanation 6**: The man is standing in front of the truck.
**Prediction**: Entail | Score 6: [0.418, 0.198, 0.364] |
| **Explanation 7**: Firetrucks and ambulances are different.
**Prediction**: Contradict | Score 7: [0.000, 0.004, 0.995] |
| **Explanation 8**: A Seattle firefighter is a man.
**Prediction**: Entail | Score 8: [0.928, 0.036, 0.027] |
| **Explanation 9**: The firefighter isn't necessarily an ambulance driver.
**Prediction**: Neutral | Score 9: [0.001, 0.552, 0.456] |

Table 9: Case study for Soft Probability Aggregation on StrategyQA Dataset. The mismatched prediction is highlighted in red.

| | |
|---|---|
| **Question**: Could every citizen of Samoa send a letter to a unique JPMorgan Chase employee?
**Ground-truth Label**: True,  **Majority Voting Prediction**: False. | |
| **Explanation 1**: Samoa has a population of about 195,000 people JPMorgan Chase has 260,000 employees.
**Prediction**: True | Score 1: [0.614, 0.373] |
| **Explanation 2**: There are 158,000 citizens of Samoa JPMorgan Chase has 247,000 employees.
**Prediction**: False | Score 2: [0.317, 0.673] |
| **Explanation 3**: There are roughly 275,000 people in Samoa JPMorgan Chase has roughly 250,000 employees Not all of JPMorgan Chase's employees are eligible to receive mail at work.
**Prediction**: False | Score 3: [0.556, 0.443] |
| **Explanation 4**: Samoa has a population of 197 751. JPMorgan Chase has about 243 000 employees worldwide. There are 50 states in the United States
**Prediction**: False | Score 4: [0.494, 0.494] |
| **Explanation 5**: There are around 194,000 people living in Samoa JPMorgan Chase employs around 250,000 people.
**Prediction**: False | Score 5: [0.526, 0.464] |
| **Explanation 6**: There are 195,308 citizens of Samoa JPMorgan Chase has 257,000 employees.
**Prediction**: True | Score 6: [0.180, 0.809] |
| **Explanation 7**: Samoa has a population of around 196,000 people JPMorgan Chase has more than 250,000 employees.
**Prediction**: False | Score 7: [0.614, 0.372] |
| **Explanation 8**: The population of Samoa is about 188 000, J P Morgan Chase employs about 250 000 people
**Prediction**: True | Score 8: [0.374, 0.616] |
| **Explanation 9**: There are 190,000 citizens of Samoa in 2017 JPMorgan Chase has over 250,000 employees. There are between 10-12 000 letters in an average day.
**Prediction**: False | Score 9: [0.465, 0.527] |

LLM predictions. By aggregating the probability score in a soft manner, we are able to rectify the final predictions, justifying the effectiveness of soft probability aggregation.

# G ADDITIONAL RESULTS ON OTHER MORE CHALLENGING TASKS

In this section, we provide additional experiments, demonstrating that EASE can also be useful to other tasks beyond those mentioned in the main experiments. For all experiments, we use LLama2-7b as the backbone LLM.

## G.1 CROSS-LINGUAL TRANSFER

We extend our framework to multi-lingual NLI tasks. For the multi-lingual NLI task, we simulate a cross-lingual setting where we provide the demonstrations and explanations in English and inference on target languages including German (DE), French (FR), Spanish (ES), and Chinese (ZH) using the examples in XNLI dataset (Conneau et al., 2018). Note that this is one common setting studied in previous multilingual reasoning benchmarks (Shi et al., 2023). We use 16-shot examples from the E-SNLI dataset as demonstrations, and for each target language, we sample 300 instances. The performance (accuracy) of our method, as well as our direct baseline (self-consistency), are shown in the below table 10.

Table 10: Performance on XNLI dataset using four target languages.

|  | DE | FR | ES | ZH | Avg. |
|---|---|---|---|---|---|
| Self-consistency | 47.15 | 44.66 | 48.33 | 39.67 | 44.95 |
| EASE | **51.50** | **48.33** | **50.66** | **43.00** | **48.37 (+7.4%)** |
| EASE w/o BLS | 48.82 | 47.33 | 49.66 | 42.00 | 46.95 |
| EASE w/o SPA | 49.49 | 42.33 | 47.00 | 40.33 | 44.79 |

From the table, we observe that our method achieves a 7.4% performance gain when compared to the self-consistency baseline, demonstrating our method can be readily applied to challenging cross-lingual scenarios.

## G.2 ARITHMETIC REASONING

For the arithmetic reasoning task, one difference is that this task requires the generation of a candidate answer in the form of a numerical value, rather than simply outputting a class or choice. Consequently, while direct adoption of soft probability aggregation may not be feasible, we can still leverage the bootstrapped LLM scorer to assign weights to different outputs generated by LLMs.

To evaluate our approach, we conducted experiments on two datasets, namely GSM8k (Cobbe et al., 2021) and SVAMP (Patel et al., 2021). For each dataset, we employed 16 examples from the training set as few-shot demonstrations. The results of our direct baseline (self-consistency) and our proposed method are summarized in the table 11.

Table 11: Performance on Arithmetic Reasoning datasets.

|  | GSM8k | SVAMP |
|---|---|---|
| Self-consistency | 22.50 | 54.33 |
| EASE (w/ BLS Only) | **24.25** | **56.33** |

From these results, it is evident that incorporating the LLM as an explanation (reasoning) scorer can further enhance the reasoning ability of the LLM.

## H ADDITIONAL RESULTS ON COMPLEXITY COMPARISON

In our extended experimentation, we sought to evaluate the comparative effectiveness of the self-consistency approach when augmented with additional samples to match the overall computational costs of our method. For this purpose, we utilized the PaLM2-S and PaLM2-L models as the backbone Large Language Models. The results are summarized in the table 12, providing a direct comparison between the standard self-consistency method and our EASE approach under similar computational constraints. Overall, we observe that EASE consistently outperforms the self-consistency approach in most scenarios, even when the latter is allocated additional computational resources.

Table 12: Comparison of Self-consistency and EASE under similar computational costs.

| LLM (→) | PaLM2-S | | | PaLM2-L | | |
|---|---|---|---|---|---|---|
| Dataset (→) | ESNLI | ECQA | StrategyQA | ESNLI | ECQA | StrategyQA |
| Self-consistency | 68.68 | 74.48 | 76.94 | 87.34 | 82.67 | 87.85 |
| Self-consistency (Same Computation) | 71.94 | 75.36 | 78.00 | 88.71 | **83.80** | 88.88 |
| EASE | **75.01** | **75.59** | **78.23** | **89.42** | 83.65 | **89.90** |

