# OpenReview forum: "Explanation-aware Soft Ensemble Empowers Large Language Model In-context Learning"
_ICLR.cc/2024/Conference — ICLR 2024 Conference Withdrawn Submission_

### Official Review · Reviewer_Bpfu · 2023-10-29

**Soundness:** 3 good
**Presentation:** 3 good
**Contribution:** 2 fair
**Rating:** 5
**Confidence:** 4

**Summary:**

This paper proposes a framework called EASE (Explanation-Aware Soft Ensemble) to improve the in-context learning capabilities of large language models (LLMs) by leveraging natural language explanations. The key ideas are:

(1) Explanation-guided ensemble: Assign a score to each prediction generated by the LLM based on the quality of its associated explanation. This is done by using another LLM as an explanation scorer. Higher quality explanations result in higher weights during ensemble. Negative explanations are generated automatically without extra annotations.

(2) Soft probability aggregation: Instead of using hard 0/1 predictions, aggregate the soft probability scores across different label verbalizers. This helps reduce noise and inconsistencies between explanations and predictions.

The experimental results on several NLU datasets show EASE outperforms baselines for in-context learning with explanations.

**Strengths:**

(1) The idea of using LLM to score explanations and weigh predictions is quite simple and straightforward.

(2) Makes good use of soft probabilities to address prediction noise and inconsistencies with explanations.

(3) Provide empirical results demonstrating consistent improvements across multiple datasets and LLMs. Ablation studies further verify the efficacy of the two proposed techniques.

(4) Well-written paper and thorough analysis and discussions.

**Weaknesses:**

(1) The approach increases computational overhead since it requires additional score for each explanation. Could be expensive for real-time applications.

(2) Relies on access to model logits which may not be available with certain blackbox LLMs, e.g. Claude-2 and ChatGPT.

(3) Improvements are not significantly, especially when models become larger.

(4) Experimental results are mainly on multi-choice/classification tasks. Not quite sure these results can hold on asthmatic reasoning tasks, e.g. GSM8K.

(5) Since the proposed techniques introduce additional computations, it will be good to know if adding more samples for self-consistency with similar overall computations can achieve similar performance. That is, adding more samples for self-consistency baselines, for fair comparisons.

**Questions:**

Missing references:

(1) Ye et al. Explanation Selection Using Unlabeled Data for Chain-of-Thought Prompting. 2023.

---

> ### Author Response · Authors · 2023-11-21
> **Initial Rebuttal to Reviewer Bpfu (Part 1)**
>
> We thank the reviewer for the constructive feedback. We discuss your raised points as follows:
>
> ***
> > W1: The approach increases computational overhead since it requires additional score for each explanation. Could be expensive for real-time applications.
>
>
> Thank you for your concern regarding the computational overhead of our approach. Indeed, our method, which incorporates soft probabilistic aggregation and a bootstrapped LLM scorer, does introduce additional computational steps. However, these steps are relatively efficient. Specifically:
>
> 1. **Decoding Time Analysis**: In our method, the additional steps — "soft probabilistic aggregation" and "Explanation Scoring" — involve generating only a single token and its probability score. This process is less time-consuming than the standard explain-then-predict step, which generates both explanations and predictions. Roughly speaking, suppose we generate N explanations for one example during the inference time, requiring $t  = N * D$ time for decoding (where D is the time for generating both explanations and predictions). Then for each explanation, suppose the decoding time for generating explanation scores and soft probability are $d_1$ and $d_2$, respectively. The additional time for explanation scoring and soft probability aggregation per explanation is $t’ = N * (d_1 + d_2)$. As has been discussed above, both  $d_1$ and $d_2$ are smaller than $D$ as it only needs to decode for one token. As a result, the additional time $t’ < 2t$.
>
> 2. **Empirical Comparison**:  In practice, our approach leads to a processing time increase of about 2 to 2.5 times compared to the self-consistency baseline. We believe this is a reasonable trade-off given the significant improvements in performance and accuracy our method provides. Furthermore, our approach has the added benefit of selecting the most relevant explanations, as shown in Figure 4, Section 4.4, which aligns better with human preferences than the baseline models.
>
> In response to your suggestion, we have conducted a more "fair" comparison with baselines that consume the same computations and observed that even with the same computations, our method EASE can still outperform the best baseline. Please refer to the **reply for W5** for detailed insights into this comparison.
>
> ***
> > W2: Relies on access to model logits which may not be available with certain blackbox LLMs, e.g. Claude-2 and ChatGPT.
>
> A: Thanks for the valuable feedback. We are well aware of the necessity of model logits in our proposed framework and have indeed recognized this limitation in **Appendix A** within our original manuscript. Besides, we have also presented a potential strategy for adapting our method to black-box LLM. This involves setting the temperature to a non-zero value, conducting multiple sampling steps, and then utilizing the frequency of the corresponding verbalizers as a proxy for the score.
>
> ***
> > W3: Improvements are not significantly, especially when models become larger.
>
> A: Thanks for the valuable feedback. It is worth noting that as the size of the model increases, it acquires more powerful reasoning abilities, and thus those standard baselines can achieve good performance (e.g. the standard in-context learning can achieve >85% performance on ESNLI datasets with PaLM2-L model, leaving less room for further improvements.). Therefore, relative improvements brought by our method may appear comparatively smaller. Nonetheless, the observed improvements are still considerable for both sizes of the backbone model (with 2.69% on large and 3.91% on small). We have also **conducted two-sided t-tests** to validate the significance of these gains and the results have been included in Tables 1 and 2. Moreover, EASE consistently outperforms the most recent state-of-the-art baselines over 7 datasets, which confirms that its improvements are generally applicable to many tasks.
>
> ***
> > W4: Experimental results are mainly on multi-choice/classification tasks. Not quite sure these results can hold on asthmatic reasoning tasks, e.g. GSM8K.
>
> A: Thank you for the suggestion. We have conducted experiments on additional datasets including 2 mathematical reasoning datasets (GSM8K, SVAMP) and a multi-lingual NLI dataset (XNLI), where we observed consistent performance gains.  We have answered this question in the general response. Please refer to the **Arithmetic Reasoning** section for details.

---

> ### Author Response · Authors · 2023-11-21
> **Initial Rebuttal to Reviewer Bpfu (Part 2)**
>
> > W5: Since the proposed techniques introduce additional computations, it will be good to know if adding more samples for self-consistency with similar overall computations can achieve similar performance. That is, adding more samples for self-consistency baselines, for fair comparisons.
>
> A: Thanks for this great advice. We have conducted an additional experiment that added more samples for self-consistency with similar overall computation costs as ours using PaLM2-S and PaLM2-L models as the LLM backbone. The experimental results are shown in the following table:
>
> | Model | | PaLM2-S  |  |   | PaLM2-L |   |
> |--|:--:|:--:|:--:|:--:|:--:|:--:|
> | Dataset  | ESNLI    | ECQA  | StrategyQA | ESNLI | ECQA  | StrategyQA |
> | Self-consistency | 68.68    | 74.48 | 76.94  | 87.34    | 82.67 | 87.85  |
> | Self-consistency (Same Computation) | 71.94    | 75.36 | 78.00  | 88.71| **83.80** | 88.88 |
> | EASE (Our method) | **75.01**   |  **75.59** | **78.23**      | **89.42**  | 83.65 | **89.90** |
>
> We observe that under the same computational cost, our method can still outperform self-consistency in most cases. We have added this result in **Appendix H** of the revised manuscript for further reference.
>
> ***
> > Q1: Missing references: Ye et al. Explanation Selection Using Unlabeled Data for Chain-of-Thought Prompting. 2023.
>
> A: Thanks for the suggestion. Actually, we already included this reference in our original manuscript. It can be found under the citation:
>
> > Xi Ye and Greg Durrett. Explanation selection using unlabeled data for in-context learning. arXiv preprint arXiv:2302.04813, 2023.
>
> The title looks slightly different because the authors changed their paper title after the ICLR submission deadline. But we have ensured that our current manuscript reflects the most recent version of their work.
>
> ***
> Thanks again for your review! We hope our responses can address your concerns. Please let us know if you have any further questions on the above issues.

---

### Official Review · Reviewer_gauf · 2023-11-03

**Soundness:** 3 good
**Presentation:** 4 excellent
**Contribution:** 4 excellent
**Rating:** 6
**Confidence:** 4

**Summary:**

This work introduces a novel method to ensemble the explanations using a bootstrapped LLM scorer and a soft probability aggregator to improve the performance of LLMs on classification tasks. The authors mainly focus on natural language understanding tasks of natural language inference and question answering, with possible extensions to generation tasks in the future work. This work builds on top of the self consistency approach, where the prediction is based on majority voting, without taking into account the variance in explanation quality and the noise introduced in the sampling stage leading to predictions which are inconsistent with the explanations. To address these issues the authors use a bootstrapped LLM scorer to score the explanations using verbalizers, prompts and in-context demonstrations of negative and positive explanations. The negative explanations are obtained without any human annotations, by assuming that explanations leading to wrong predictions are in fact negative. Finally these scores are aggregated using the sum of the probabilities associated with each classification label and weighted by the explanation score, thereby reducing the effect of the noise originating from the sampling based algorithms and obtaining a more accurate result. The experiments are conducted on several benchmarks like E-SNLI, ECQA etc. Compared to baselines like in-context learning, predict then explain framework, self-consistency etc, the EASE framework introduced in this work performs the best on all the benchmarks using PALM as the backbone as well as other open-sourced models like FLAN-UL2 and Llama-2. This shows the effectiveness and the generalisability of the proposed method

**Strengths:**

This work produces a simple intuitive approach to improve the prediction of LLMs using the explanations generated. The experiments conducted on various benchmarks and different LLMs shows that the proposed method can be generalised to any problem. All the in-context demonstrations required for the bootstrap LLM scorer are generated automatically without the need for human annotations. The ablations performed shows the effectiveness of using both the bootstrapped LLM scorer and the soft probability aggregations introduced in this work. The study of the inconsistency ratio and their corresponding relation with the improvement in the performance across several benchmarks, thoroughly explains the need for the soft probability aggregation framework. Similarly the need for bootstrapped LLM scorer is explained using human annotations and compared with the previous methods of lexical match and NLI models, showcasing its superior performance. Overall, there is a very detailed explanation of the introduced method and the ablations performed motivating the need for the proposed framework.

**Weaknesses:**

There is a limited number of types of tasks studied, confined only to NLI and QA in the English language. There is a possibility of the framework failing in cases where the LLM scorer is wrong at every stage of the framework starting with the bootstrapped LLM scorer assigning a higher weight to the explanation being a positive verbaliser when in fact it could be negative. This error could be carried downstream to the soft probability aggregator where the prediction generated using the explanation could be wrong, leading to a worse performance. Since this method is tested out on popular easy tasks like NLI this effect may not be visible. If the framework could be tested on other tasks like multilingual NLI or uncommon sense reasoning where the LLMs do not have a particularly good performance, this effect may be visible.

**Questions:**

1)Were the EP and the PE methods tested using other sampling methods or other greedy sampling, which may have given diverse explanations leading to better results?

2)Are there any insights into why the performance of the EASE framework without bootstrapped LLM scorer works slightly better/on par with the EASE framework for certain benchmarks?

---

> ### Author Response · Authors · 2023-11-21
> **Initial Rebuttal to Reviewer gauf (Part 1)**
>
> We thank the reviewer for the valuable suggestions for improvements. For your several questions, we clarify as follows:
>
> ***
> > W1: There is a limited number of types of tasks studied, confined only to NLI and QA in the English language. There is a possibility of the framework failing in cases where the LLM scorer is wrong at every stage of the framework starting with the bootstrapped LLM scorer assigning a higher weight to the explanation being a positive verbaliser when in fact it could be negative. This error could be carried downstream to the soft probability aggregator where the prediction generated using the explanation could be wrong, leading to a worse performance. Since this method is tested out on popular easy tasks like NLI this effect may not be visible. If the framework could be tested on other tasks like multilingual NLI or uncommon sense reasoning where the LLMs do not have a particularly good performance, this effect may be visible.
>
> A: Thanks for pointing out this issue. To demonstrate that our proposed techniques can be readily applied to the scenarios where the performance of vanilla LLM in-context learning (self-consistency) is not good enough, we would like to first point out that for ANLI-R1, R2, R3 datasets using LLaMa2-7b as the LLM backbone, the self-consistency approach only achieves a notably low accuracy range of 36%-40%. It's noteworthy that our method (EASE) still manages to attain performance gains ranging from 1.5% to 5.2% for these datasets. This indicates that our method remains effective even when the initial results from the LLM are suboptimal, which demonstrates the robustness and effectiveness of our EASE framework for improving performance in challenging scenarios.
>
> Besides, per your suggestion, we have run additional experiments on **multilingual NLI and arithmetic reasoning tasks** using the LLama2-7b as the backbone LLM to support our claims. Please refer to our general response for details.
>
> ***
> > Q1: Were the EP and the PE methods tested using other sampling methods or other greedy sampling, which may have given diverse explanations leading to better results?
>
> A: Thank you for your insightful query regarding the testing of EP and PE methods with different sampling strategies. We addressed this question as follows:
>
> 1. **EP and PE Methods Original Approach**: For both the EP and PE methods, we stick to the original methodology outlined in papers [1,2]: the LLM only generates one response and thus no prediction aggregation. We employ a greedy sampling approach (setting temperature t=0) for both EP and PE methods.
> 2. **Temperature Sampling for Diverse Explanations**: When generating multiple explanations and predictions for aggregation, setting the temperature t=0 is not meaningful (as the explanations and predictions will always be the same). Thus, we use the temperature sampling with temperature=0.7 to generate diverse explanations.
> 3. **EP Method with Temperature Sampling**: For the EP method, incorporating temperature sampling is equal to the self-consistency baseline [3], which we have reported the number in both Table 1 and Table 2.
> 4. **PE Method with Temperature Sampling**: Regarding the PE method, we indeed attempted to integrate the temperature sampling techniques with t=0.7 as mentioned above to generate diverse explanations. However, it was observed that this approach had a **negative** effect on the final performance. We present the performance results across various tasks using LLaMa2-7b as follows:
>
>
> |Model | ESNLI | ANLI1 | ANLI2 | ANLI3 | ECQA |  StrategyQA | OpenbookQA |
> |---|:--:|:--:|:--:|:--:|:--:|:--:|:--:|
> | PE | 54.25 |  37.83 |  37.50 | 34.37 |  52.33 |  56.21 |  56.48 |
> | PE + Sampling | 50.50 | 34.50 | 35.33 | 33.66 | 51.66 | 54.45 | 55.60 |
>
> The potential reason behind this unexpected outcome could be that, unlike the EP pipeline, the PE pipeline may not benefit significantly from the generated explanations. In addition, the sampling step introduces additional noise into the prediction process. As a result, diverse explanations do not appear to provide a clear advantage for the PE pipeline. We have provided more discussions about this issue in **Section 4.2**.

---

> ### Author Response · Authors · 2023-11-21
> **Initial Rebuttal to Reviewer gauf (Part 2)**
>
> ***
> > Q2: Are there any insights into why the performance of the EASE framework without bootstrapped LLM scorer works slightly better/on par with the EASE framework for certain benchmarks?
>
> A: Thank you for this insightful question. To explain this result, it's important to acknowledge that the few-shot in-context learning performance for LLM often exhibits instability and sensitivity to the choice of input examples [4]. The table below demonstrates the performance on five random splits of the ANLI dataset using PaLM2-S as the backbone:
>
> |         | EASE  | EASE w/o BLS | EASE w/o SPA |
> |---|:--:|:--:|:--:|
> | Split 1 | **65.33** | 63.67        | 64.67        |
> | Split 2 | 60.66 | 65.67        | **66.67**        |
> | Split 3 | 65.33 | **66.67**        | 63.67        |
> | Split 4 | **70.33** | 68.33        | 69.00       |
> | Split 5 | **70.66** | 69.67        | 69.67        |
>
> From the table, it is evident that our proposed EASE framework achieves the best performance on three out of five splits. However, EASE's performance may suffer when the quality of the few-shot demonstrations is suboptimal, which results in a slightly lower average performance compared to our variants on certain datasets. We are committed to providing an unbiased evaluation and avoid cherry-picking results in this process.
>
> We have also conducted similar assessments on other datasets, and we have included the number of cases where EASE outperforms its two variants. For instance, "4/5" means our method achieves the best performance in four out of five random splits. The results show that our method consistently achieves the best performance across most data splits:
>
> |   Backbone LLM      | ESNLI | ANLI1 | ANLI2 | ANLI3 | ECQA | StrategyQA | OpenbookQA |
> |----|:--:|:--:|:--:|:--:|:--:|:--:|:--:|
> | PaLM2-S | 4/5   | 3/5   | 3/5   | 3/5   | 4/5  | 3/5  | 4/5   |
> | PaLM2-L | 4/5   | 3/5   | 3/5   | 4/5   | 3/5  | 3/5   | 3/5  |
>
> Furthermore, it's important to note that while the performance gain of this component may not be as significant on relatively easy datasets, it becomes more evident on more challenging datasets, such as multi-lingual NLI and arithmetic reasoning tasks, as presented in the general response.
>
> In response to your suggestion, we have expanded our discussion of this issue in the "Limitation Section" in **Appendix A** to provide a more comprehensive understanding.
>
>
> >[1] Wei et al. "Chain-of-thought prompting elicits reasoning in large language models." NeurIPS 2022.
> >
> >[2] Lampinen et al. "Can language models learn from explanations in context?." Findings of EMNLP 2022.
> >
> >[3] Wang et al. "Self-Consistency Improves Chain of Thought Reasoning in Language Models." ICLR 2023.
> >
> >[4] Min et al. "MetaICL: Learning to Learn In Context." NAACL 2022.
>
> ***
> Thank you again for your review! We hope our responses can address your concerns. Please let us know if you have any further questions.

---

### Official Review · Reviewer_HfbG · 2023-11-03

**Soundness:** 2 fair
**Presentation:** 2 fair
**Contribution:** 2 fair
**Rating:** 5
**Confidence:** 4

**Summary:**

The paper proposes a framework called EASE (Explanation-Aware Soft Ensemble) to improve in-context learning for large language models (LLMs) using natural language explanations. The authors address the limitations of existing methods by introducing explanation-aware ensemble and soft probability aggregation techniques. They conduct experiments on seven natural language understanding tasks and four LLMs, demonstrating the effectiveness of their proposed framework.

**Strengths:**

- The authors provide extensive experimental results on multiple datasets and LLMs, demonstrating the effectiveness of their proposed framework. They compare their approach with several baselines and achieve superior performance in most cases.

**Weaknesses:**

- The paper introduces a framework that leverages explanations to enhance in-context learning for LLMs, which is not quite novel as there are existing works that uses explanations to enhance in-context learning for LLMs as well [1,2].

- While the experimental results are promising, the paper lacks a theoretical analysis of the proposed framework. A more in-depth analysis of the underlying principles and theoretical guarantees would strengthen the paper.

- The authors briefly mention the limitations of their approach, such as the reliance on LLMs for explanation scoring and the absence of negative explanations. However, they do not provide a thorough discussion of these limitations or potential ways to mitigate them.

- The authors compare their full framework with several baselines, but they do not provide ablation studies to analyze the individual contributions of the explanation-aware ensemble and soft probability aggregation techniques. A more detailed analysis of these techniques would provide a better understanding of their impact.


References

- [1] Krishna, Satyapriya, et al. "Post hoc explanations of language models can improve language models." arXiv preprint arXiv:2305.11426 (2023).

- [2] Bills, Steven, et al. "Language models can explain neurons in language models." URL https://openaipublic. blob. core. windows. net/neuron-explainer/paper/index. html.(Date accessed: 14.05. 2023) (2023).

**Questions:**

None

---

> ### Author Response · Authors · 2023-11-21
> **Initial Rebuttal to Reviewer HfbG (Part 1)**
>
> We thank the reviewer for the useful feedback. We have answered your questions in the response below.
>
> ***
> > W1: The paper introduces a framework that leverages explanations to enhance in-context learning for LLMs, which is not quite novel as there are existing works that uses explanations to enhance in-context learning for LLMs as well [1,2].
> >
> >[1] Krishna, Satyapriya, et al. "Post hoc explanations of language models can improve language models." arXiv preprint arXiv:2305.11426 (2023).
> >
> >[2] Bills, Steven, et al. "Language models can explain neurons in language models." URL https://openaipublic. blob. core. windows. net/neuron-explainer/paper/index. html.(Date accessed: 14.05. 2023) (2023).
>
> A: Thank you for pointing out these references. We have added more discussions about two works in the Section 2 “Related Work” (c.f. our new draft). While they are somewhat relevant to our study from the perspective of LLM interpretability, we want to emphasize the differences between these papers and our work:
>
> - Paper [1] generates rationales by **extracting** top-k works with the highest attention score, which is an extractive-based approach and relies on additional attention score information. In contrast, our work focuses on the **generation** of free-text rationales by Large Language Models.
>
> - Paper [2] studies interpretability at the **neuron** level to explain the factors in text that trigger a neuron's activation. On the other hand, our study will generate **instance-level** explanations using Large Language Models.
>
> Additionally, we would like to highlight that our method stands apart from previous works as it introduces a novel approach for effectively harnessing explanations with Large Language Models for in-context learning. Our experiment results further substantiate the performance gains over previous methodologies.
>
> ***
> > W2: While the experimental results are promising, the paper lacks a theoretical analysis of the proposed framework. A more in-depth analysis of the underlying principles and theoretical guarantees would strengthen the paper.
>
> A: We would like to point out that in this work, we focus more on the empirical study with a detailed analysis of the effect of different modules (Sections 4.5-4.6). While we acknowledge the significance of theoretical analysis for understanding model performance, it's important to note that making a formal derivation from a theoretical perspective, especially for Large Language Models with billions of parameters, presents substantial challenges. We admit that this paper focuses on the empirical studies of using explanations to assist LLM in-context learning,  while theoretical analysis is an important avenue of future work. The field of deep learning theory, particularly in the context of stochastic non-convex optimization, often requires strong assumptions to facilitate formal derivations. We believe that a comprehensive theoretical analysis of this topic warrants a publication of its own, given its complexity and significance.
>
> ***
> > W3: The authors briefly mention the limitations of their approach, such as the reliance on LLMs for explanation scoring and the absence of negative explanations. However, they do not provide a thorough discussion of these limitations or potential ways to mitigate them.
>
> A: Thanks for the suggestion. We would like to highlight that both issues of **LLMs for explanation scoring** and **the absence of negative explanations** have been already discussed and potential solutions are also provided in **Section 3.3 “Explanation-guided Ensemble”**. Specifically, we delve into the utilization of LLMs for explanation scoring, as well as the generation of negative explanations, which are detailed in the **LLM as Explanation Scorer** and **Bootstrapped LLM Scorer** components, respectively.
>
> For other limitations, such as the reliance on logit scores, we have also provided solutions in **Appendix A** due to the page limit. We invite you to refer to Appendix A for further details.

---

> ### Author Response · Authors · 2023-11-21
> **Initial Rebuttal to Reviewer HfbG (Part 2)**
>
> > W4: The authors compare their full framework with several baselines, but they do not provide ablation studies to analyze the individual contributions of the explanation-aware ensemble and soft probability aggregation techniques. A more detailed analysis of these techniques would provide a better understanding of their impact.
>
> A: Thank you for your suggestion. It's crucial to emphasize that our original manuscript already includes ablation studies for both the **explanation-aware ensemble** and **soft probability aggregation** techniques. In **Section 4.2** and **Section 4.3**, we have presented the experimental results of EASE and its two variants across seven datasets and four LLMs, all detailed in **Table 1** and **Table 2**. These variants include EASE without the Bootstrapped LLM Scorer, the component responsible for achieving the explanation-aware ensemble, and EASE without soft probability aggregation, denoted as "EASE w/o BLS" and "EASE w/o SPA," respectively.
>
> Most importantly, we kindly point out that we have conducted comprehensive studies on different alternatives, including other existing techniques for the **explanation-aware ensemble** technique in **Section 4.4** and the **soft probability aggregation** technique in **Section 4.5**. These sections include a thorough analysis of the efficacy of various metrics and strategies used in each component.
>
> ***
> Thanks again for your review! We hope our response has solved your concerns. Feel free to let us know if you have any further questions.

---

### Author Response · Authors · 2023-11-21
**General Response to All Reviewers**

To further justify the effectiveness of our proposed method that it can work on tasks under more challenging settings where **the LLM may not perform well initially**, we have run additional experiments on **multilingual NLI and arithmetic reasoning** tasks using the LLaMa2-7b as the backbone LLM to support our claims. The details are described as follows:

***
### **Multilingual NLI**

For the multilingual NLI task, we adopt the cross-lingual setting where we provide the demonstrations and explanations in English and inference on target languages including German (DE), French (FR), Spanish (ES), and Chinese (ZH) using the examples in XNLI dataset [1]. Note that this is one common setting studied in previous multilingual reasoning benchmarks [2]. We use 16-shot examples from the E-SNLI dataset as demonstrations, and for each target language, we sample 300 instances. The performance (accuracy) of our method, as well as our best performing baseline (self-consistency), are shown in the below table:

|| DE | FR | ES | ZH | Avg. |
|:--:|:--:|:--:|:--:|:--:|--|
| Self-consistency | 47.15 | 44.66 | 48.33 | 39.67 | 44.95 |
| Ours | **51.50** | **48.33** | **50.66** | **43.00** | **48.37**  (+7.4%) |
| Ours w/o BLS | 48.82 |47.33 | 49.66 | 42.00 | 46.95 |
| Ours w/o SPA | 49.49 | 42.33 | 47.00 | 40.33 | 44.79 |

From the table, we observe that our method achieves an average 7.4% performance gain when compared to the self-consistency baseline.

***
### **Arithmetic Reasoning**

For the arithmetic reasoning task, we would like to point out that this task requires the generation of a candidate answer in the form of a numerical value, rather than simply outputting a class or choice. Consequently, while direct adoption of soft probability aggregation may not be feasible, we can still leverage the bootstrapped LLM scorer to assign weights to different outputs generated by LLMs.

To evaluate our approach, we conducted experiments on two datasets, namely GSM8k [3] and SVAMP [4]. For each dataset, we employed 16 examples from the training set as few-shot demonstrations. The results of our best performing baseline (self-consistency) and our proposed method are summarized in the table below:

|| GSM8k | SVAMP |
|:--:|:--:|:--:|
| Self-consistency | 22.50 | 54.33  |
| Ours (w/ BLS) | **24.25** | **56.33**  |

From these results, it is evident that incorporating the LLM as an explanation (reasoning) scorer can further enhance the reasoning ability of the LLM. We have added these results in Appendix G of the modified manuscript. We hope that these additional experiments address the concerns raised by the reviewers and provide further support for the efficacy of our approach.


>[1] Conneau et al. "XNLI: Evaluating Cross-lingual Sentence Representations." EMNLP 2018.
>
>[2] Shi et al. "Language models are multilingual chain-of-thought reasoners." ICLR 2023.
>
>[3] Cobbe et al. "Training verifiers to solve math word problems." arXiv preprint arXiv:2110.14168 (2021).
>
>[4] Patel et al.  "Are NLP Models really able to Solve Simple Math Word Problems?." NAACL 2021.

---

### Author Response · Authors · 2023-11-21
**Summary of Revisions**

In response to valuable feedback from reviewers, we have revised our manuscript as requested, denoting these changes with $\textbf{\color{blue}blue}$ highlights. While answering detailed questions of each review, we also list a summary of major updates:

1.  We have added more discussions on two related works mentioned by the reviewer HfbG and updated the paper title information for Ye et al. mentioned by the reviewer Bpfu.
2. We have added additional experimental results on XNLI (multi-lingual NLI), GSM8k & SVAMP (mathematical reasoning) tasks in Appendix G, as requested by the reviewer gauf and Bpfu.
3. We have added a performance comparison of our method and the strongest baseline under the same computations for a fair comparison in Appendix H, as requested by reviewer Bpfu.

We hope that the revised manuscript will address the concerns raised by the reviewers.

---

### Author Response · Authors · 2023-11-21
**A Gentle Reminder**

Dear All Reviewers,

Thank you again for your valuable feedback! We would like to kindly remind the reviewers that the author/reviewer discussion phase **ends tomorrow**, on November 22nd. We hope that our rebuttals and additional experiments have addressed the reviewers' concerns and improved paper's overall quality. If there are any additional comments you would like to provide, please don't hesitate to share them. We look forward to engaging in a constructive discussion during the rebuttal phase.

Best,

Authors